# Deep learning-based classification of speech disorder in stroke and hearing impairment

**Joo Kyung Park[1], Sae Byeol Mun[2], Young Jae Kim[3], Kwang Gi Kim [1,2,3]***

1 Department of Biomedical Engineering, College of Medicine, Gachon University, Gil Medical Center, Incheon, Republic of Korea, 2 Department of Health Sciences and Technology, Gachon Advanced Institute for Health Sciences & Technology, Gachon University, Incheon, Republic of Korea, 3 Gachon Biomedical & Convergence Institute, Gachon University Gil Medical Center, Incheon, Republic of Korea

* kimkg@gachon.ac.kr

## Abstract

### Background and objective

Speech disorders can arise from various causes, including congenital conditions, neurological damage, diseases, and other disorders. Traditionally, medical professionals have used changes in voice to diagnose the underlying causes of these disorders. With the advancement of artificial intelligence (AI), new possibilities have emerged in this field. However, most existing studies primarily focus on comparing voice data between normal individuals and those with speech disorders. Research that classifies the causes of these disorders within the abnormal voice data, attributing them to specific etiologies, remains limited. Therefore, our objective was to classify the specific causes of speech disorders from voice data resulting from various conditions, such as stroke and hearing impairments (HI).

### Methods

We experimentally developed a deep learning model to analyze Korean speech disorder voice data caused by stroke and HI. Our goal was to classify the disorders caused by these specific conditions. To achieve effective classification, we employed the ResNet-18, Inception V3, and SEResNeXt-18 models for feature extraction and training processes.

### Results

The models demonstrated promising results, with area under the curve (AUC) values of 0.839 for ResNet-18, 0.913 for Inception V3, and 0.906 for SEResNeXt-18, respectively.

### Conclusions

These outcomes suggest the feasibility of using AI to efficiently classify the origins of speech disorders through the analysis of voice data.

**Data availability statement:** All data underlying the findings described in this study are fully available without restriction. The dataset used in this study has been deposited in the AIHub database, a publicly accessible repository. Repository Name: AIHub Dataset Link: https://www.aihub.or.kr/aihubdata/data/view.do?aihubDataSe=data&dataSetSn=71434 To access the data, users must register for a free account on AIHub. For researchers located outside of South Korea, accessing the data requires a separate agreement with the Korea Information Society Agency (NIA) according to the AIHub data usage policy. All relevant data, including the data points behind the reported means and variance measures, can be downloaded from this repository.

**Funding:** This work was supported by the Small and Medium Business Technology Innovation Development Project (Market Expansion) Support Project (Grant No: S3383835) funded by the Ministry of SMEs and Startups (MSS, Korea). Dr. Kwang Gi Kim received funding from the Ministry of SMEs and Startups (MSS, Korea). The funders had no role in study design, data collection and analysis, decision to publish, or preparation of the manuscript.

**Competing interests:** The authors have declared that no competing interests exist.

## Introduction

Speech disorders significantly impair communication abilities, drastically diminishing the quality of life of affected individuals [1–3]. These disorders can stem from various causes, including congenital conditions, neurological damage, diseases, or other disorders [4–10]. Consequently, medical professionals have traditionally used vocal changes to diagnose the origins of speech disorders. Initial assessments relied on the auditory judgments of examiners; however, advancements in science and technology have enabled more precise acoustical analysis [11,12]. This progress has enhanced the quantification and accuracy of diagnoses, although limitations remain in fully understanding and interpreting complex voice signals. Currently, various studies have leveraged the advantages of AI for diagnosing causes through acoustical analysis of voices. AI has demonstrated the potential to deliver diagnoses that are more accurate than those made by medical specialists or to support specialists in making more informed diagnostic decisions [13–15].

In 2023, Kim et al. extracted features using Mel-frequency cepstral coefficients (MFCC) from the voice data of 230 healthy individuals and 230 stroke patients and applied them to a convolutional neural network (CNN), achieving an accuracy of 99.60% [16]. In 2020, Singh and Xu extracted features using MFCC from the voice data of 860 healthy individuals and 140 Parkinson's patients, applying the data to a support vector machine (SVM) and achieving an accuracy of 99.00% [17]. In 2018, Wu et al. achieved an accuracy of 77.00% by extracting features via the short-time Fourier transform (STFT) from the voice data of 482 healthy individuals and 482 patients with various speech disorders and applying them to a 2D-CNN [18]. Using both 2D-CNN and long short-term memory (LSTM) approaches, Syed et al. compared and researched the application of features extracted using MFCC on voice data from 1600 healthy individuals and 400 patients experiencing various speech disorders, achieving an accuracy of 97.11% using 2D-CNN [19]. Shih et al. extracted features using STFT and MFCC and applied them to convolutional neural network-gated recurrent units (CNN-GRU), CNN, and LSTM for comparison; they achieved an accuracy of 98.38% using CNN-GRU [20].

Despite the attainment of high accuracy rates, existing research is limited. Most studies are based on small datasets and primarily focus on comparing voice data between normal individuals and those experiencing speech disorders. Consequently, classification tasks have often focused on a single cause among the various origins of speech disorders, such as laryngeal cancer or Parkinson's disease.

"This study aims to overcome these limitations by utilizing speech disorder data collected through scripts from patients with stroke, the representative neurological disease, and HI, to classify whether the speech disorders are caused by stroke or HI using a CNN-based algorithm."

## Materials and methods

### Experimental setup

The research environment for this study was established using a system equipped with two NVIDIA RTX 2080 Ti GPUs (NVIDIA, Santa Clara, CA, USA), 329 GB RAM,

and a 22-core Intel(R) Xeon(R) Gold 6238 CPU @ 2.10 GHz (Intel, Santa Clara, CA, USA). The system operated on the Ubuntu LTS operating system (version 16.04.7) within a Python environment (version 3.6.10). Data preprocessing was conducted using the Librosa (version 0.9.2) and Pandas (version 1.1.5) libraries, while model training was performed using the TensorFlow library (version 2.6.2). Additionally, statistical analysis and evaluation were conducted using libraries such as Sklearn (version 0.24.2), Imblearn (version 0.8.1), and Matplotlib (version 3.3.4), alongside the R environment (version 4.3.3), utilizing the pROC package (version 1.18.5) for Receiver Operating Characteristic (ROC) curve analysis.

## Data

This study was designed as a retrospective analysis using pre-existing clinical voice data collected from patients with stroke and hearing impairment at Ewha Womans University Seoul Hospital. The institutional review board of Ewha Womans University Seoul Hospital (IRB Number: SEUMC 2022-03-012) approved this study; the requirement of informed consent was waived due to the retrospective nature of the study design. The data collection targeted 250 patients with stroke and 250 patients with HI, both representing potential congenital or acquired causes of speech disorders. Detailed information on the patients' gender, age, diagnosis, intelligibility, and degree of disability is summarized in Table 1.

Recordings were made in a lossless format to ensure high quality. In collaboration with medical personnel and HI specialists, patients were provided with 702 types of diverse scripts that one might encounter in daily life. These scripts were presented in full sentences to maintain the continuous characteristics of speech and were recorded, resulting in a total of 2,000 voice data sets. Each recording was assigned a ground truth label of either stroke or hearing impairment (HI) based on clinical diagnoses confirmed by certified neurologists and audiologists at Ewha Womans University Seoul Hospital. To ensure the reliability of these labels, an independent review was conducted by speech-language pathologists who cross-checked the diagnostic information with the observed speech characteristics, such as nasalization and consonant weakening for stroke patients and syllable segmentation for HI patients. This multi-step verification process minimized labeling errors and ensured that each voice sample accurately represented its corresponding disorder category. The recordings were collected in soundproof speech therapy rooms and underwent a comprehensive data review, cleansing, and processing procedure. Particularly, considering the speech characteristics of severely disabled individuals, voice data were cleansed and processed in collaboration with speech therapists based on the International Phonetic Alphabet. The scripts used for the recordings are listed in Table 2.

Waveforms for comparison between voice data are depicted in Fig 1. In the case of individuals with speech disorders, two notable characteristics are observed: (1) nasalization, where the airflow from the vocal tract resonates simultaneously in the nasal and oral cavities, causing syllables to merge and sound muffled; and (2) consonant weakening, where the boundaries between consonants and vowels become unclear. Additionally, in the case of HI, the characteristic of syllable segmentation is more pronounced compared with stroke patients.

The recorded set of 2,000 voice data was divided into training, validation, and testing datasets at a ratio of 8:1:1, resulting in 1,600 data points for the training dataset, 200 for the validation dataset, and 200 for the testing dataset.

## Data preprocessing

As illustrated in Fig 2, data preprocessing began with the resampling of all voice data to 16 kHz, in accordance with the Nyquist–Shannon sampling theorem. To ensure consistency in data handling, all voice recordings were uniformly segmented to a length of 4 s. Recordings shorter than 2 s were excluded from the analysis, and recordings between 2 and 4 s were extended to a uniform length using zero padding, thereby enhancing signal stability, frequency transformation, and resolution. Additionally, to address variations in the distance between the recorder and the patient during recording sessions, voice signals were normalized to a range of -1–1. Subsequently, signals were re-normalized to a range of 0–1 to optimize model training efficiency. Through dataset preprocessing, we ultimately secured 2,914 data points for training, 382 for validation, and 378 for testing purposes.

**Table 1. Characteristics of data on gender, age, diagnosis, intelligibility, and degree of disability for stroke and Hearing-Impaired (HI) patients.**

| Data information | | Count | Ratio |
|---|---|---|---|
| **Gender** | Male | 265 | 53.00 |
| | Female | 235 | 47.00 |
| **Age** | 10–20 | 131 | 26.20 |
| | 30–40 | 124 | 24.80 |
| | 50–60 | 155 | 31.0 |
| | Over 70 | 90 | 18.00 |
| **Diagnosis** | Stroke | 1000 | 50.00 |
| | HI* | 1000 | 50.00 |
| **Intelligibility** | SIR1** | 106 | 5.30 |
| | SIR2 | 151 | 7.55 |
| | SIR3 | 331 | 16.55 |
| | SIR4 | 695 | 34.75 |
| | SIR5 | 717 | 35.85 |
| **Degree of disability** | Mild | 840 | 42.00 |
| | Moderate | 239 | 11.95 |
| | Moderately | 100 | 5.00 |
| | Severe | 268 | 13.40 |
| | Profound | 553 | 27.65 |

*HI, Hearing Impairment.

**SIR, Speech Intelligibility Rating.

**Table 2. List of Diverse Scripts Used for One-Sentence Unit Voice Recordings.**

| ID | Content | Count |
|---|---|---|
| 1 | "코클리어 엔 세븐 블루투스 연결 방법 알려 줘." ("Please show me how to connect the Cochlear N7 with Bluetooth.") | 2 |
| 2 | "펜션 근처 지하철역 조회 부탁합니다." ("I'd like to know the nearest subway station to a rental cottage.") | 7 |
| 3 | "만성 중이염 검색해 줘." ("Search for chronic otitis media for me. ") | 3 |
| 4 | "삼십 번 좌석 예약을 그만둘게. " ("I'll cancel the reservation for seat thirty.") | 2 |
| 5 | "즐겨찾기 메뉴를 켜줘. " ("Please open the bookmarks.") | 2 |
| ⋮ | ⋮ | ⋮ |
| 698 | "생수 두 병 부탁해요. " ("Could you bring me two bottles of water?") | 3 |
| 699 | "지금 날씨는 좀 나아졌는지 알아봐 줘. " ("Check if the weather has improved.") | 4 |
| 700 | "오늘 비 소식 있어? " ("Is there any rain forecast for today?") | 1 |
| 701 | "탁상 취침 등 꺼. " ("Turn off the bedside lamp.") | 7 |
| 702 | "오늘 태풍 온다는 것 같았는데 맞아? " ("Was there supposed to be a typhoon today?") | 2 |

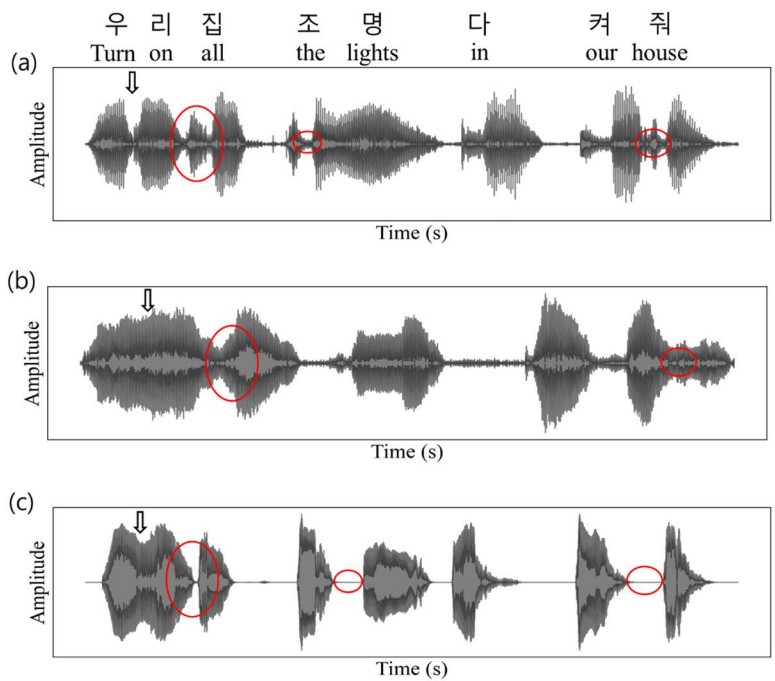

**Fig 1. Data Waveform Samples.** (a) Normal Speech Waveform, (b) Stroke Patient's Waveform, (c) Hearing-Impaired (HI) Patient's Waveform.

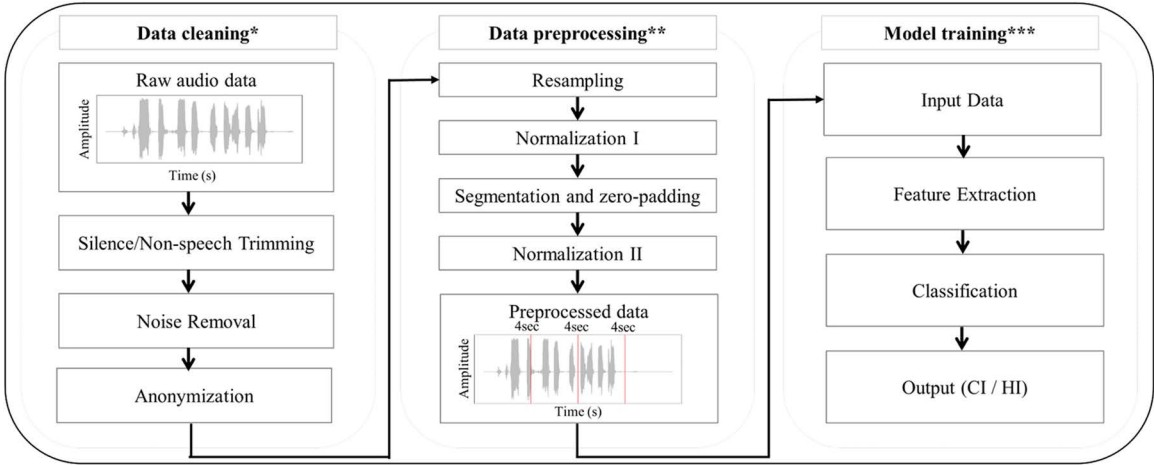

**Fig 2. Flow chart for feature extraction and classification of stroke and HI using speech data.** * Data cleaning performed using Audacity ** Resampling to 16,000Hz; Normalization to a range of -1 to 1; Segmentation into 4s intervals with zero-padding, resulting in 3,674 segments, each containing 64,000 samples; Normalization to a range of 0 to 1 applied to each segment; Creation of a single NumPy file for training *** 1D-CNN for data-driven feature extraction and training using ResNet-18, Inception V3, and SEResNeXt-18 models.

## Models for classifying the causes of speech disorder using voice data

Feature extraction, as shown in Fig 2, was performed using a 1D-CNN model by applying convolution operations to voice data. This study adopted the 1D-CNN approach due to existing research indicating the superior performance of 1D-CNNs over methods utilizing recurrent neural networks (RNN) such as LSTM, MFCC, and 2D-CNNs that use spectrograms for voice data processing [21,22].

The model incorporated three architectures: ResNet, SEResNeXt, and Inception. The ResNet model employed by He et al. in 2016 addressed the vanishing gradient problem in deep neural networks through residual connections [23]. As illustrated by Szegedy et al., the Inception network enables the efficient extraction of spatial information at multiple levels using filters of various dimensions [24]. A set of transformations with the same topology defines the building blocks of the ResNeXt architecture, a modular variation of ResNet, which implements a multi-branch architecture based on an additional dimension known as cardinality [25]. The SEResNeXt architecture is built upon ResNeXt with the integration of squeeze-and-excitation (SE) blocks, enhancing the network's representational power through dynamic channel-wise feature recalibration [26].

Hyperparameters were set with a batch size of 16 and a learning rate of 0.0001. The ReduceLROnPlateau algorithm was applied to optimize performance by gradually decreasing the learning rate if the model plateaued during training. The Adam optimizer was used to train the model over 200 epochs, and an early stopping algorithm using the binary cross-entropy loss function was introduced to prevent overfitting.

## Results

In this study, the performance of the trained models was validated against a separately constructed test dataset. The evaluation metrics employed were sensitivity, specificity, and accuracy, calculated based on the true positive (TP), false positive (FP), true negative (TN), and false negative (FN) ratios from the confusion matrix. Additionally, AUC was used as a performance metric derived from the ROC curve. Delong's test was utilized to verify the statistical significance of performance differences between the models. A comparative analysis among the three models was conducted based on the versions that exhibited the best performance.

As shown in Table 3, all models demonstrated excellent performance across all evaluation metrics. In terms of sensitivity, ResNet-18 recorded a performance of 77.53, Inception V3 93.26, and SEResNeXt-18 87.64. ResNet-18 exhibited a specificity of 82.00, Inception V3 84.50, and SEResNeXt-18 84.00. ResNet-18 demonstrated an accuracy of 79.89, Inception V3 88.62, and SEResNeXt-18 85.71. Overall, based on the evaluation metrics derived from the confusion matrix, Inception V3 demonstrated relatively high performance among the three architectures.

Fig 3 presents the confusion matrices of each model, illustrating the detailed classification performance for stroke and hearing-impaired (HI) patients. It shows how each model performed in terms of true positive, true negative, false positive, and false negative counts for each class. Notably, Inception V3 in Fig 3b has significantly fewer false negatives, with 12 cases, compared to 40 in ResNet-18 in Fig 3a and 22 in SEResNeXt-18 in Fig 3c. Additionally, Inception V3 exhibits fewer false positives, with 31 cases, compared to 36 in ResNet-18 and a similar count of 32 in SEResNeXt-18. The comparison of the confusion matrices provides an additional layer of insight into the classification behavior of each model, highlighting how they handle classification errors across different classes.

**Table 3. Performance Metrics of a Binary Classifier for Differentiating Between Stroke and Hearing-Impaired (HI) Patients Using Speech Disorder Data.**

| Model | Sensitivity (95% CI) | Specificity (95% CI) | Accuracy (95% CI) | AUC (95% CI)* |
|---|---|---|---|---|
| ResNet-18 | 77.53±8.01 (71.40~83.66) | 82.00±2.01 (76.68~87.32) | 79.89±2.94 (75.85~83.93) | 0.839±0.038 (0.800~0.881) |
| Inception V3 | 93.26±4.47 (89.57~96.94) | 84.50±2.50 (79.48~89.52) | 88.62±0.70 (85.42~91.83) | 0.913±0.019 (0.881~0.947) |
| SEResNeXt-18 | 87.64±5.09 (82.81~92.48) | 84.00±2.92 (78.92~89.08) | 85.71±2.34 (82.19~89.24) | 0.906±0.009 (0.872~0.939) |

*The 95% confidence intervals for AUC were estimated via the bootstrap method.

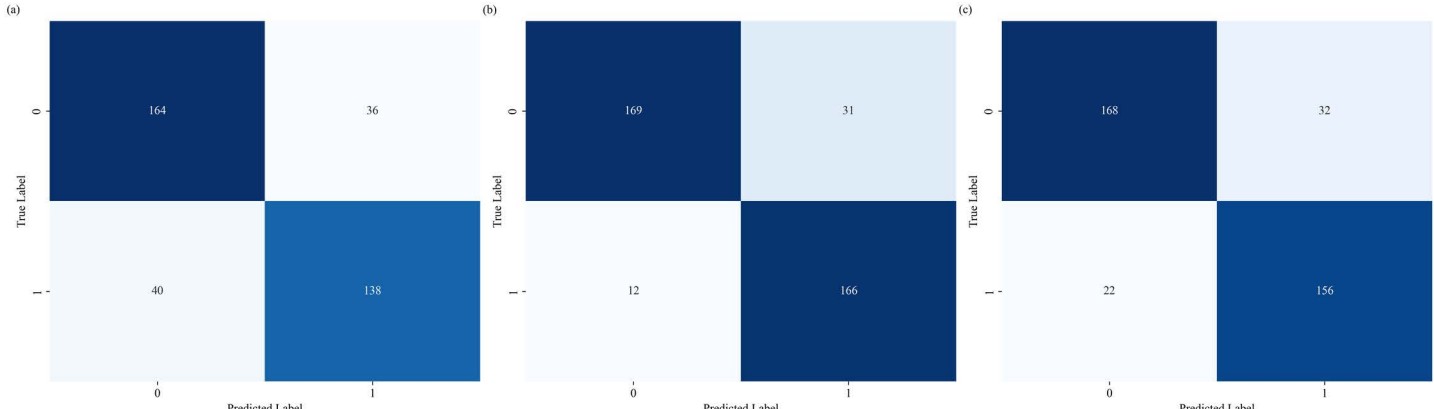

**Fig 3. Confusion Matrices Comparison of Deep Learning Models for Classifying Stroke and HI Disorders.** (a) ResNet-18, (b) Inception V3, (c) SEResNeXt-18.

Fig 4 shows a comparison of performance in terms of AUC, confirming that the models effectively classify both stroke and HI. ResNet-18 recorded an AUC of 0.839 with a standard deviation of 0.038 and a 95% confidence interval of 0.800–0.881, demonstrating relatively lower figures compared with the other two models. Inception V3 demonstrated the highest performance with an AUC of 0.913, a standard deviation of 0.019, and a 95% confidence interval of 0.881–0.947. Among all three models, SEResNeXt-18 exhibited a moderate performance with an AUC of 0.906, a standard deviation of 0.009, and a 95% confidence interval of 0.872–0.939.

As shown in Table 4, the comparison of AUC between models through Delong's test revealed that the performance difference between ResNet-18 and the other two models was statistically significant, with $p < 0.05$. However, the performance difference between Inception V3 and SEResNeXt-18 was not statistically significant, as indicated by $p < 0.227$.

## Discussion

This paper presents a study on classifying stroke and HI using speech disorder voice data. We acquired high-quality source data and engaged in data cleansing and processing, which included effective noise removal. Subsequently, we performed data preprocessing steps such as normalization, zero-padding, and segmentation. Utilizing a 1D convolution layer, we automatically extracted and learned the local characteristics of complex waveform patterns from raw voice data. This model-based approach afforded remarkable classification performance, with accuracy scores of 79.89 for ResNet-18, 88.62 for Inception V3, and 85.71 for SEResNeXt-18. The AUC scores of 0.839 for ResNet-18, 0.913 for Inception V3, and 0.906 for SEResNeXt-18 confirm the models' effective differentiation between speech disorders caused by stroke and HI. However, despite all models demonstrating strong performance, there were notable differences in their classification accuracy. These differences can be attributed to the distinct architectural characteristics of each model. ResNet-18, while benefiting from residual connections that mitigate the vanishing gradient problem and enable deeper network training, is primarily optimized for simplicity and computational efficiency [23]. This results in an inability to effectively capture subtle, multi-scale features in complex datasets such as speech disorder data, which requires a more nuanced approach to feature extraction. Consequently, ResNet-18's lower performance may be linked to its reliance on standard convolutional filters that do not adapt as dynamically to varying spatial patterns, making it less suited for highly variable speech signals. Similarly, SEResNeXt-18, which incorporates Squeeze-and-Excitation (SE) blocks to adaptively recalibrate feature maps and enhance sensitivity to important features, showed improved performance over ResNet-18 but still fell short compared to Inception V3. While SEResNeXt-18 excels in focusing on critical features by adjusting the significance of channel-wise information, its relatively narrower network structure and limited ability to analyze complex patterns at multiple scales may

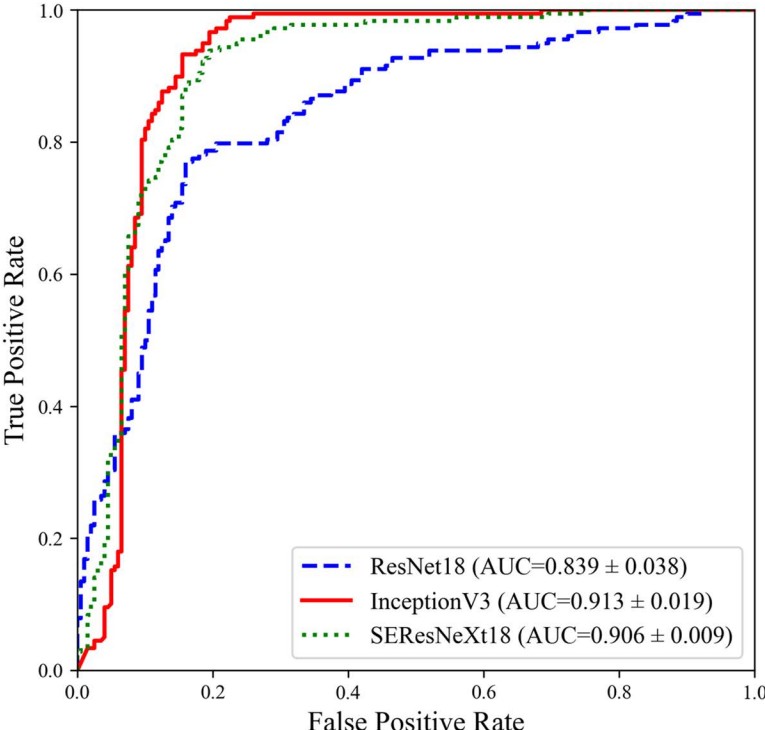

**Fig 4. ROC Curve Comparison of Models Classifying Stroke and HI from Speech Disorder Data.**

**Table 4. Statistical Significance of Performance Differences Among Models as Determined by Pairwise Delong's Test P-value Comparisons in Terms of AUC.**

| Model Comparison | P-Value† | | |
|---|---|---|---|
| | ResNet-18 | Inception V3 | SEResNeXt-18 |
| **ResNet-18** | | P < 0.05* | P < 0.05* |
| **Inception V3** | | | P = 0.227 |
| **SEResNeXt-18** | | | |

†*p*-value based on pairwise comparison of ROC curves.

*Results were considered statistically significant at *p*-value < 0.05.

hinder its performance when faced with highly heterogeneous speech patterns. The SE block's recalibration advantage is effective, but it may not be sufficient for capturing complex temporal dynamics in voice data [25,26]. In contrast, Inception V3's superior performance can be attributed to its distinctive multi-branch architecture, which utilizes multiple convolutional filter sizes within a single layer to capture diverse spatial hierarchies. This design allows Inception V3 to learn both fine-grained and coarse features simultaneously, making it particularly well-suited for the intricate and multi-scale nature of speech disorder data. Moreover, Inception V3 employs factorized convolutions and auxiliary classifiers, which help reduce computational complexity while preserving its capacity to analyze deep feature representations. This combination of deep feature extraction and efficient computational strategies enables the model to maintain high classification accuracy even in the presence of subtle variations in speech patterns caused by stroke and HI [24]. Consequently, Inception V3's ability to balance depth, width, and multi-scale feature extraction makes it the most effective model for the given task, demonstrating

not only the highest overall accuracy but also the most balanced performance in differentiating between stroke and HI speech disorder classes. In addition to each model's structural characteristics, the number of parameters likely also contributed to the observed performance differences. Within the constraints of our experimental environment, this study selected the highest-performing variant from each model family—ResNet-18, Inception V3, and SEResNeXt-18—for comparison across models. ResNet-18 has 925,697 parameters, Inception V3 has 12,345,761, and SEResNeXt-18 has 1,308,033, with Inception V3 containing the highest parameter count among the models. In the AUC comparison, Delong's test results showed a statistically significant difference between ResNet-18 and the other two models ($p < 0.05$), while no significant difference was observed between Inception V3 and SEResNeXt-18 ($p < 0.227$). These findings further support the suitability of Inception V3's balanced architecture and higher parameter count for complex speech classification tasks [27].

This research is significant for utilizing a larger dataset compared with those used in previous studies and achieving excellent performance across various evaluation metrics. It excels in accurately classifying speech disorder voices caused by stroke and HI rather than merely distinguishing between normal and abnormal speakers. Unlike previous research that primarily focused on the phonation of the vowel/a/, this study utilized entire scripts, allowing for a more comprehensive analysis of speech patterns and characteristics such as pitch breaks, voice fatigue, and voice breaks, which is expected to enhance accuracy and flexibility in screening diagnoses [28,29].

However, this study is limited to Korean voice data and focuses only on stroke and HI among various causes of speech disorders, which indicates a lack of diversity in the data. Toward the development of multi-classification or cause-specific classification models, future research should aim to build datasets suitable for various languages and cultures and secure extensive voice datasets for diverse causes inducing speech disorders, such as amyotrophic lateral sclerosis, Parkinson's disease, Down syndrome, and multiple sclerosis. This paper presents significant results in experimentally classifying the causes of speech disorders stemming from stroke and HI using speech disorder voice data. The developed classification models can be employed for the early detection and diagnosis of speech disorders, potentially mitigating symptom severity and enhancing the effectiveness of treatments. In particular, the high sensitivity and balanced performance of these models make them well-suited for identifying early signs of speech deterioration, even when symptoms are subtle and not easily detectable by the human ear. This capability is crucial in clinical practice, where early intervention is often the key to preventing further progression of speech disorders. For instance, in stroke rehabilitation, prompt detection of dysarthria or other related symptoms can help tailor personalized speech therapy programs, thereby improving recovery outcomes. Moreover, these models can serve as efficient tools for remote monitoring and risk assessment. By leveraging automated voice analysis through digital devices, clinicians can track patients' speech changes over time and adjust treatment plans accordingly without requiring frequent hospital visits. This approach not only reduces the burden on healthcare facilities but also improves patient accessibility to specialized care, especially for those in rural or medically underserved areas. Additionally, in contexts such as pandemics or when mobility is restricted, remote voice monitoring provides a safe and effective way to ensure continuous health management and intervention.

The advancement of AI technology and its integration with the medical field are expected to play a significant role in assisting medical diagnoses and improving accuracy and speed, leading to the realization of enhanced medical services. As demonstrated in this study, machine learning models can contribute to more nuanced diagnostic capabilities, potentially extending beyond speech disorders to other areas where precise pattern recognition is crucial. This evolution in diagnostic methodology could ultimately pave the way for AI-assisted systems that support clinicians in making more informed decisions, thereby optimizing patient outcomes and revolutionizing healthcare delivery.

## Author contributions

**Conceptualization:** Sae Byeol Mun.

**Data curation:** Sae Byeol Mun.

**Formal analysis:** Joo Kyung Park.

**Funding acquisition:** Kwang Gi Kim.

**Investigation:** Sae Byeol Mun, Young Jae Kim.

**Methodology:** Joo Kyung Park, Sae Byeol Mun.

**Project administration:** Young Jae Kim.

**Supervision:** Kwang Gi Kim.

**Validation:** Joo Kyung Park, Young Jae Kim, Kwang Gi Kim.

**Visualization:** Joo Kyung Park.

**Writing – original draft:** Joo Kyung Park.

**Writing – review & editing:** Young Jae Kim, Kwang Gi Kim.

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
