## [Decision Letter · Decision Letter 0]

3 Oct 2024

PONE-D-24-28326Deep learning-based classification of speech disorder in stroke and hearing impairment: An analysis of neural network performance and clinical implicationsPLOS ONE

Dear Dr. Kim,

Thank you for submitting your manuscript to PLOS ONE. After careful consideration, we feel that it has merit but does not fully meet PLOS ONE’s publication criteria as it currently stands. Therefore, we invite you to submit a revised version of the manuscript that addresses the points raised during the review process.

I agree with the reviewers about the need for a revision of the manuscript.

We look forward to receiving your revised manuscript.

Kind regards,

Diego A. Forero, MD; PhD

Academic Editor

PLOS ONE

“This work was supported by the Technology Innovation Program(or Industrial Strategic Technology Development Program(K_G012001185601, Building Data Sets for Artificial Intelligence Learning) funded By the Ministry of Trade Industry & Energy(MOTIE, Korea), and by the GRRC program of Gyeonggi province.  [GRRC-Gachon2023(B01), Development of AI-based medical imaging technology]”

5. Please amend either the title on the online submission form (via Edit Submission) or the title in the manuscript so that they are identical.

Reviewers' comments:

Reviewer's Responses to Questions

**Comments to the Author**

1. Is the manuscript technically sound, and do the data support the conclusions?

Reviewer #1: Yes

Reviewer #2: Yes

2. Has the statistical analysis been performed appropriately and rigorously? 

Reviewer #1: Yes

Reviewer #2: Yes

3. Have the authors made all data underlying the findings in their manuscript fully available?

Reviewer #1: No

Reviewer #2: Yes

4. Is the manuscript presented in an intelligible fashion and written in standard English?

Reviewer #1: Yes

Reviewer #2: Yes

5. Review Comments to the Author

Reviewer #1: This is a well-written manuscript about using deep learning to classify speech disorders in individuals with stroke and hearing impairment. I have some suggestions to improve this manuscript.

Materials and Methods

Please add details about the ground truth for this study.

Results

I believe this study would be more compelling if the authors included the confusion matrix results for each model.

Discussion

Please include a discussion on why each model achieved different accuracy levels in classifying this type of data.

Additionally, expand on the clinical applications of this system in practice.

Reviewer #2: Introduction

The authors highlight the need for this work by presenting the advantages of developing classification-based artificial intelligence models to support the diagnosis of speech disorders. The theoretical foundation of existing studies is outlined, along with the present gaps in the research.

The objective of the study is clearly stated and aligns with the title of the work.

Materials and Methods

It is recommended to specify the type of study conducted to clarify for the reader that it is retrospective.

The specifications of the equipment used are provided, as well as the Python libraries and statistical analysis tools employed.

In addition, the characteristics of the audio recordings are detailed, along with the approval by the Seoul Hospital board regarding the use of these primary source recordings.

The characteristics of the population and sample size are presented.

The procedure for recording the audio samples is described, including the details of the scripts used and the physical conditions during the recordings. A flow diagram is included to enhance the reader's comprehension.

Furthermore, the inclusion and exclusion criteria for the recordings, as well as the total number of recordings used, are outlined. The 8:1:1 ratio used for model development, testing, and validation in the machine learning system is also presented.

The methodological rigor in the development of the AI model, as well as the data selection and classification processes, is evident.

Results

The statistical results of the models are presented, demonstrating very good performance under the evaluated conditions. One model outperformed the other two presented.

Relevant statistical techniques are used to compare the models and analyze data classification.

Discussion

The discussion emphasizes important aspects such as the use of a larger dataset compared to previous studies. Accurate classification of speech disorders associated with stroke and hearing impairment is achieved, rather than merely distinguishing between normal and abnormal speakers. The analysis is based on complete scripts, allowing for a more comprehensive examination of speech patterns and characteristics, contrasting with previous studies that focused on phonation of the vowel /a/.

The discussion also highlights the advantages of analyzing features like pitch breaks, vocal fatigue, and voice breaks, which are expected to enhance the accuracy and flexibility of diagnoses.

The study's limitations primarily concern its focus on voice data in Korean and its restriction to stroke and hearing impairment, without considering other speech disorders.

6. PLOS authors have the option to publish the peer review history of their article (what does this mean? ). If published, this will include your full peer review and any attached files.

**Do you want your identity to be public for this peer review?** For information about this choice, including consent withdrawal, please see our Privacy Policy .

Reviewer #1: No

Reviewer #2: No

---

## [Author Response · Author response to Decision Letter 1]

8 Nov 2024

Dear Editor and Reviewers,

We sincerely thank you for your thoughtful and constructive comments on our manuscript. We have carefully addressed each point raised in your reviews and made the necessary revisions to improve the manuscript. Below is a detailed point-by-point response to the reviewers' comments, outlining the changes made in the revised version of the manuscript.

Thank you again for your valuable feedback, which has significantly contributed to enhancing the quality of our work.

Response: We have carefully ensured that the manuscript meets PLOS ONE's style requirements, including file naming conventions

Response: We acknowledge the guidelines on code sharing and will follow the recommended practices for code availability as outlined.

“This work was supported by the Technology Innovation Program(or Industrial Strategic Technology Development Program(K_G012001185601, Building Data Sets for Artificial Intelligence Learning) funded By the Ministry of Trade Industry & Energy(MOTIE, Korea), and by the GRRC program of Gyeonggi province. [GRRC-Gachon2023(B01), Development of AI-based medical imaging technology]”

Response: We have included the statement, 'The funders had no role in study design, data collection and analysis, decision to publish, or preparation of the manuscript,' in the cover letter as requested.

Response: We confirm our intention to make the data freely accessible upon acceptance of the manuscript.

5. Please amend either the title on the online submission form (via Edit Submission) or the title in the manuscript so that they are identical.

Response: The title has been amended to be consistent across the manuscript and submission form.

Response: We have reviewed the reference list to ensure it is complete and correct, and there are no retracted papers cited.

Reviewers' comments:

Reviewer's Responses to Questions

Comments to the Author

3. Have the authors made all data underlying the findings in their manuscript fully available?

Reviewer #1: No

Response: The data underlying the results presented in the study are available from the AIHub database (https://www.aihub.or.kr). To access the data, users must register for a free account on AIHub. All relevant data, including the data points behind the reported means and variance measures, can be downloaded from this repository without restrictions. However, for researchers located outside of South Korea, accessing the data requires a separate agreement with the Korea Information Society Agency (NIA) as per the AIHub data usage policy.

Reviewer #2: Yes

5. Review Comments to the Author

Reviewer #1: This is a well-written manuscript about using deep learning to classify speech disorders in individuals with stroke and hearing impairment. I have some suggestions to improve this manuscript.

Response: We sincerely appreciate your positive evaluation and valuable feedback.

Materials and Methods

Please add details about the ground truth for this study.

Response: We have expanded the "Materials and Methods" section to include detailed information on the ground truth. We specified how the ground truth labels were assigned based on clinical diagnoses confirmed by certified neurologists and audiologists, and described the multi-step verification process by independent speech-language pathologists to ensure label accuracy.

Results

I believe this study would be more compelling if the authors included the confusion matrix results for each model.

Response: We have added confusion matrix results for each model in the Results section.

Discussion

Please include a discussion on why each model achieved different accuracy levels in classifying this type of data.

Response: We have expanded the Discussion section to provide a detailed analysis of why each model performed differently, highlighting the architectural strengths and limitations of ResNet-18, SEResNeXt-18, and Inception V3 in the context of speech disorder classification. Additionally, we have included a comparison of parameter counts across models, as this factor also likely contributed to the observed performance differences.

Additionally, expand on the clinical applications of this system in practice.

Response: We have added a section in the Discussion that outlines potential clinical applications of the proposed classification system, emphasizing its use for early diagnosis, rehabilitation monitoring, and remote risk assessment for stroke and hearing impairment patients.

Reviewer #2: Introduction

The authors highlight the need for this work by presenting the advantages of developing classification-based artificial intelligence models to support the diagnosis of speech disorders. The theoretical foundation of existing studies is outlined, along with the present gaps in the research.

The objective of the study is clearly stated and aligns with the title of the work.

Response: We sincerely appreciate your positive evaluation and valuable feedback.

Materials and Methods

It is recommended to specify the type of study conducted to clarify for the reader that it is retrospective.

Response: We have revised the "Materials and Methods" section to specify that this study was designed as a retrospective analysis using pre-existing clinical voice data.

The specifications of the equipment used are provided, as well as the Python libraries and statistical analysis tools employed.

In addition, the characteristics of the audio recordings are detailed, along with the approval by the Seoul Hospital board regarding the use of these primary source recordings.

The characteristics of the population and sample size are presented.

The procedure for recording the audio samples is described, including the details of the scripts used and the physical conditions during the recordings. A flow diagram is included to enhance the reader's comprehension.

Furthermore, the inclusion and exclusion criteria for the recordings, as well as the total number of recordings used, are outlined. The 8:1:1 ratio used for model development, testing, and validation in the machine learning system is also presented.

The methodological rigor in the development of the AI model, as well as the data selection and classification processes, is evident.

Results

The statistical results of the models are presented, demonstrating very good performance under the evaluated conditions. One model outperformed the other two presented.

Relevant statistical techniques are used to compare the models and analyze data classification.

Discussion

The discussion emphasizes important aspects such as the use of a larger dataset compared to previous studies. Accurate classification of speech disorders associated with stroke and hearing impairment is achieved, rather than merely distinguishing between normal and abnormal speakers. The analysis is based on complete scripts, allowing for a more comprehensive examination of speech patterns and characteristics, contrasting with previous studies that focused on phonation of the vowel /a/.

The discussion also highlights the advantages of analyzing features like pitch breaks, vocal fatigue, and voice breaks, which are expected to enhance the accuracy and flexibility of diagnoses.

The study's limitations primarily concern its focus on voice data in Korean and its restriction to stroke and hearing impairment, without considering other speech disorders.

Response: We sincerely thank you for your thoughtful and encouraging assessment and feedback.

---

## [Decision Letter · Decision Letter 1]

25 Nov 2024

Deep learning-based classification of speech disorder in stroke and hearing impairment

PONE-D-24-28326R1

Dear Dr. Kim,

We’re pleased to inform you that your manuscript has been judged scientifically suitable for publication and will be formally accepted for publication once it meets all outstanding technical requirements.

Kind regards,

Diego A. Forero, MD; PhD

Academic Editor

PLOS ONE

Additional Editor Comments (optional):

Reviewers' comments:

Reviewer's Responses to Questions

**Comments to the Author**

1. If the authors have adequately addressed your comments raised in a previous round of review and you feel that this manuscript is now acceptable for publication, you may indicate that here to bypass the “Comments to the Author” section, enter your conflict of interest statement in the “Confidential to Editor” section, and submit your "Accept" recommendation.

Reviewer #1: All comments have been addressed

2. Is the manuscript technically sound, and do the data support the conclusions?

Reviewer #1: Yes

3. Has the statistical analysis been performed appropriately and rigorously? 

Reviewer #1: Yes

4. Have the authors made all data underlying the findings in their manuscript fully available?

Reviewer #1: Yes

5. Is the manuscript presented in an intelligible fashion and written in standard English?

Reviewer #1: Yes

6. Review Comments to the Author

Reviewer #1: (No Response)

7. PLOS authors have the option to publish the peer review history of their article (what does this mean? ). If published, this will include your full peer review and any attached files.

**Do you want your identity to be public for this peer review?** For information about this choice, including consent withdrawal, please see our Privacy Policy .

Reviewer #1: No

---

## [Editor Report · Acceptance letter]

PONE-D-24-28326R1

PLOS ONE

Dear Dr. Kim,

I'm pleased to inform you that your manuscript has been deemed suitable for publication in PLOS ONE. Congratulations! Your manuscript is now being handed over to our production team.

Kind regards,

on behalf of

Dr. Diego A. Forero

Academic Editor

PLOS ONE